# Supplementary UV-B Radiation Effects on Photosynthetic Characteristics and Important Secondary Metabolites in *Eucommia ulmoides* Leaves

**DOI:** 10.3390/ijms24098168

**Published:** 2023-05-03

**Authors:** Siqiu Xiao, Dewen Li, Zhonghua Tang, Hongling Wei, Ying Zhang, Jing Yang, Chunjian Zhao, Ying Liu, Wei Wang

**Affiliations:** 1College of Chemistry, Chemical Engineering and Resource Utilization, Northeast Forestry University, Harbin 150040, China; 2Key Laboratory of Forest Plant Ecology, Ministry of Education, Northeast Forestry University, Harbin 150040, China; 3Engineering Research Center of Forest Bio-Preparation, Ministry of Education, Northeast Forestry University, Harbin 150040, China; 4Institute of Advance Carbon Conversion Technology, Huaqiao University, Xiamen 361021, China

**Keywords:** *Eucommia ulmoides* Oliver, supplementary UV-B radiation, physiological characteristics, secondary metabolites, transcriptome

## Abstract

To explore the effects of ultraviolet light supplementation on the photosynthetic characteristics and content of secondary metabolites in the leaves of *Eucommia ulmoides* Oliver (*E. ulmoides*), the effects of supplementary UV-B (sUV-B) radiation on the medicinally active components of *E. ulmoides* were comprehensively evaluated. In our study, we selected leaves of five-year-old *E. ulmoides* seedlings as experimental materials and studied the effect of supplemental ultraviolet-B (sUV-B) radiation on growth, photosynthetic parameters, photosynthetic pigments, fluorescence parameters, and secondary metabolites of *E. ulmoides* using multivariate analysis. The results showed that the leaf area and the number of branches increased after sUV-B radiation, which indicated that sUV-B radiation was beneficial to the growth of *E. ulmoides*. The contents of chlorophyll a and chlorophyll b increased by 2.25% and 4.25%, respectively; the net photosynthetic rate increased by 5.17%; the transpiration rate decreased by 35.32%; the actual photosynthetic efficiency increased by 10.64%; the content of the secondary metabolite genipin increased by 12.9%; and the content of chlorogenic acid increased by 75.03%. To identify the genes that may be related to the effects of sUV-B radiation on the growth and development of *E. ulmoides* leaves and important secondary metabolites, six cDNA libraries were prepared from natural sunlight radiation and sUV-B radiation in *E. ulmoides* leaves. Comparative analysis of both transcriptome databases revealed a total of 3698 differential expression genes (DEGs), including 1826 up-regulated and 1872 down-regulated genes. According to the KOG database, the up-regulated unigenes were mainly involved in signal transduction mechanisms [T] and cell wall/membrane biogenesis [M]. It is also involved in plant hormone signal transduction and phenylpropanoid biosynthesis metabolic pathways by the KEGG pathway, which might further affect the physiological indices and the content of chlorogenic acid, a secondary metabolite of *E. ulmoides*. Furthermore, 10 candidate unigenes were randomly selected to examine gene expression using qRT-PCR, and the six libraries exhibited differential expression and were identical to those obtained by sequencing. Thus, the data in this study were helpful in clarifying the reasons for leaf growth after sUV-B radiation. And it was beneficial to improve the active components and utilization rate of *E. ulmoides* after sUV-B radiation.

## 1. Introduction

*Eucommia ulmoides* Oliver *(E. ulmoides)*, also known as bakelite, belongs to the Eucommiaceae and is one of the most precious woody Chinese herbal medicines in China. It was listed as a national second-class endangered species protection plant in 1999 [1]. Modern pharmacological studies reveal the remarkable therapeutic effects of *E. ulmoides* on hypertension, hyperglycemia, diabetes, obesity, osteoporosis, Parkinson’s disease, Alzheimer’s disease, aging, and sexual dysfunction [2]. Most of the medicinally active components in *E. ulmoides* are secondary metabolites [1], and these active components have great application prospects in medicine. The aucubin, genipin, geniposide, and geniposidic acid isolated from *E. ulmoides* all belong to iridoid [3], and chlorogenic acid belongs to phenolic compounds, with the highest content in *E. ulmoides* [4]. Increasing the content of secondary metabolites of *E. ulmoides* is expected to improve its medicinal active components, which will help it play a greater role in medical health and clinical application.

The effect of UV-B radiation (280–320 nm) on plants is between that of UV-C radiation and UV-A radiation, which has the functions of disinfecting and sterilizing, inducing plants to develop tolerance to diseases, activating plant immune function, altering plant physiological activities, inhibiting plant overgrowth, and increasing chlorophyll content in leaves [5,6]. Experimental studies have shown that proper sUV-B radiation could increase leaf area, improve the robustness index, enhance leaf antioxidant enzyme activity, and improve plant disease resistance, providing a basis for further investigation of the mechanism of UV-B radiation effects on plant growth [7]. Most plants have a UV barrier, which can protect them, but it cannot always fully protect them. Low-dose UV-B radiation can trigger the expression of plant secondary metabolite genes, improve the medicinally active components of plants, and promote the development of rhizomes [8]. A high dose of UV-B radiation will cause programmed cell death, change the proportion of plant tissues, change the wax layer protecting leaves, dwarf plants, shorten the branch pitch, inhibit flowering, and even whiten plant leaves so that photosynthesis cannot be carried out. Different UV treatment conditions have different effects on plants. Therefore, in order to improve the quality of products and promote their growth and development, the dosage added during UV-B radiation needs to be appropriate [9]. However, the effects of sUV-B radiation on medicinal plants, especially the secondary metabolites associated with important medicinal plants, are rarely studied.

Therefore, five-year-old *E. ulmoides* seedlings were used as the experimental material, and low-dose UV-B radiation (2.81 KJ·m^−2^·d^−1^) was added to ordinary LED fill light. Combined with actual production requirements, the growth index, photosynthesis, and secondary metabolites of leaves were determined, and the effects of sUV-B radiation on them in *E. ulmoides* were explored by transcriptome analysis.

## 2. Results

### 2.1. Leaf Anatomy Analysis

Anatomical observation of leaf cross-sections indicated that the epidermal cells were larger than the control (Table 1 and Figure 1). Compared to the control, the palisade tissue became long and loose, the spaces between cells increased, and the cells of the spongy tissue deformed and became dispersed (Figure 1). The palisade tissue cells of the leaf after sUV-B treatment were 2.5 times larger than those of the control. These results indicate that the anatomy of the cells in the leaves changed significantly after sUV-B treatment.

### 2.2. Effect of sUV-B Radiation on the Growth Indexes of E. ulmoides

As shown in Table 2, compared with CK, the leaf width, leaf length, leaf area, and leaf thickness of *E. ulmoides* increased after sUV-B radiation and increased, respectively, by 39.57%, 37.56%, 98.36%, and 34.78%, of which leaf length, leaf area, and leaf thickness were significantly different (*p* < 0.05). The number of branches of *E. ulmoides* after sUV-B radiation increased; the number of branches increased by 1.9 times compared to CK, and there was a significant difference (*p* < 0.05). The fresh and dry weight of leaves increased after UV-B radiation, but the relative water content decreased by 0.03 times, and there was no significant difference (*p* > 0.05). The correlations among the individual indicators were increased, and the response of *E. ulmoides* to sUV-B radiation was not the performance of a single trait, and each indicator had different roles in the construction of the evaluation system of the sUV-B radiation ability of *E. ulmoides*.

### 2.3. Effects of sUV-B Radiation on Photosynthesis of E. ulmoides

The results of the photosynthetic indexes between natural light radiation and sUV-B radiation of *E. ulmoides* are shown in Table 3 and Table 4. Compared with CK, chlorophyll a and chlorophyll b contents were increased, and the difference was not significant (*p* > 0.05), while the ratio of chlorophyll a to chlorophyll b and carotenoid content were reduced and showed significant differences (*p* < 0.05). After sUV-B radiation, the net photosynthetic rate and transpiration rate of *E. ulmoides* leaves were higher than CK, but the stomatal conductance and intercellular CO_2_ concentration were lower in the T treatment group than in CK. The net photosynthetic rate, transpiration rate, stomatal conductance, and intercellular CO_2_ concentration were not significantly different (*p* > 0.05) between the CK group and the T group.

The maximum photosynthetic efficiency Fv/Fm values were lower than CK, and the difference between CK and Tr treatment groups was not significant (*p* > 0.05). The actual photosynthetic efficiency was opposite to the maximum photosynthetic efficiency, and the T group was higher than the CK group, and there was a significant difference (*p* < 0.05).

### 2.4. Changes in the Content of Secondary Metabolites of E. ulmoides

As shown in Table 5, the effects of sUV-B radiation on the contents of aucubin, chlorogenic acid, genipin, geniposide, and geniposidic acid were all different compared with CK. sUV-B radiation had a significant elevating effect on chlorogenic acid and genipin contents, which were 1.74 and 1.12 times higher, respectively, than CK, and there was a significant (*p* < 0.05) induction of accumulation of chlorogenic acid contents after sUV-B radiation. And there was no significant (*p* > 0.05) difference in the accumulation of genipin content. The content of geniposide, geniposidic acid, and aucubin decreased after sUV-B radiation, respectively, by 77.03%, 96.01%, and 93.19%, and all of them were significantly different (*p* < 0.05).

### 2.5. Correlation Analysis

Significant differences were found between physiological indicators, photosynthetic characteristics indicators, and secondary metabolite contents of plants after sUV-B radiation (Figure 2). For example, leaf number showed significant differences between all parameters except for no significant differences with leaf dry weight, and showed positive correlations with carotenoids, chlorophyll ratio, stomatal conductance, aucubin, and geniposide, and positive correlations with genipin, chlorogenic acid, moisture content, chlorophyll, and the number of branches showed a negative correlation. There was a significant positive difference between secondary metabolite content and each parameter and a strong positive correlation between chlorogenic acid content and leaf width, leaf length, leaf area, leaf thickness, branch number, leaf fresh weight, moisture content, chlorophyll, transpiration rate, and actual photosynthetic efficiency (Figure 2).

### 2.6. Denovo Assembly and Sequence Annotation

The clean reads were assembled into 62,287 transcripts and 42,333 unigenes using Trinity, with a total length of 70,449,627 bp and 41,056,860 bp, a mean length of 1131 bp and 969 bp, and an N50 of 1762 bp and 1582 bp, respectively (Table 6). The quality of the whole sequencing was high and thus adequate for follow-up analyses. Most (38,709) of the transcripts ranged from 200 to 1000 bp, accounting for 62% of the total transcripts; 22% (13,533) of the transcripts ranged from 1100 to 1900 bp; and 10,045 (16%) of the transcripts were longer than 2000 bp. Most (29,473) of the unigenes ranged from 201 to 1000 bp, accounting for 70% of the total unigenes; 18% (7700) of the unigenes ranged from 1100 to 1900 bp; and 5159 (12%) of the unigenes were longer than 2000 bp. The differentially expressed genes (DEGs) were screened, and an MA map was generated to display the level of expression of all genes between groups (FPKM) and the distribution of DEGs. Differential gene cluster analysis showed that there were significant differences in gene expression patterns between the CK and the sUV-B treatments.

### 2.7. Sequence Annotation and Classification

After aligning the unigene sequences to protein databases, 42,333 unigenes were annotated. The numbers and percentage of unigenes annotated within the SwissProt, TrEMBL, nonredundant (NR), Pfam, Cluster of Orthologous Groups for Eukaryotic Complete Genomes (KOG), Gene Ontology (GO), and Kyoto Encyclopedia of Genes and Genomes (KEGG) Orthology (KO) databases were 14,988 (35.40%), 24,272 (57.30%), 24,500 (57.90%), 19,708 (46.60%), 19,523 (46.10%), 19,282 (45.5%), and 6790 (16%), respectively (Table 7).

### 2.8. Functional Classification of Differentially Expressed Genes

Figure 3a shows that the up-regulated genes involved in translation, ribosomal structure, and biogenesis [J] and transcription [K] are among the 19 categories in information storage and processing; in cellular processes and signaling, the up-regulated genes involved in posttranslational modification, protein turnover, and chaperones [V]; defense mechanism and signal transduction mechanisms [T]; in metabolism, the up-regulated genes involved in energy production and conversion [C]; carbohydrate transport and metabolism [G] and inorganic ion transport and metabolism [P]. Meantime, in information storage and processing, the down-regulated genes involved in translation, ribosomal structure, and biogenesis [J] and replication, recombination, and repair [L]; in cellular processes and signaling, the down-regulated genes involved in posttranslational modification, protein turnover, and chaperones [O]; in metabolism, the down-regulated genes involved in carbohydrate transport and metabolism [G]; energy production and conversion [C]; and amino acid transport and metabolism [E].

We studied the complex biological behaviors of genes in more detail and obtained pathway annotations for unigenes. In total, 6790 unigenes were annotated in the KEGG database and were assigned to 128 pathways. In total, the up-regulated genes were enriched for photosynthesis, phenylpropanoid biosynthesis, plant hormone signal transduction, starch and sucrose metabolism, nitrogen metabolism, ABC transporters, oxidative phosphorylation, nicotinate and nicotinamide metabolism, and cyanoamino acid metabolism. Compared with the up-regulated gene, the pathway of the down-regulated gene was significantly less enriched than that of the up-regulated gene, only enriched in 4 metabolic pathways: ribosome, flavonoid biosynthesis, carbon fixation in photosynthetic organisms, and one carbon pool by folate (Figure 3b). The up-regulated genes are enriched in photosynthesis in the KEGG database, phenylpropanoid biosynthesis, plant hormone signal transduction, starch and sucrose metabolism, nitrogen metabolism, ABC transporters, oxidative phosphorylation, nicotinate and nicotinamide metabolism, and cyanoamino acid metabolism.

With annotations in the NR database, 19,282 unigenes were assigned to GO categories with 4734 unique functional terms. Of all the unigenes assigned GO categories, 16,830 unigenes were shared by the three categories, whereas 1571, 2351, and 812 unigenes were uniquely assigned to the GO categories of biological processes, cellular components, and molecular functions, respectively (Figure 3c). In summary, the upregulated genes were mainly involved in cell wall/membrane biogenesis and plant hormone signal transduction after sUV treatment, which might promote plant metabolism and accelerates plant growth.

### 2.9. DEGs Related to Plant Auxin and Phenylpropanoid Biosynthesis

According to the results of KOG and KEGG, DEGs related to the photosynthesis pathway, plant auxin, and phenylpropanoid biosynthesis pathways (associated with cell wall biogenesis) were analyzed. In plant hormone signal transduction, the related genes from the auxin response protein aux/iaa, auxin-responsive protein Iaa, and Saur family protein were upregulated (Figure 4), which could promote the elongation and growth of plant cells and accelerate plant growth. The high expression of CRE1 (EC: 2.7.13.3) is involved in zeatin biosynthesis metabolism, which could also promote cell division and then accelerate plant growth.

In phenylpropanoid biosynthesis, peroxidase [EC:1.11.1.7], beta-glucosidase [EC:3.2.1.21], and caffeic acid 3-O-methyltransferase [EC:2.1.1.68] were upregulated, involved mostly in the biosynthesis of guaiacyl lignin and 5-hydroxy-guaiacyl lignin (Figure 5).

### 2.10. Validation and Expression Analyses of Key Enzyme Genes

To validate the changes in gene expression patterns, 10 candidate unigenes associated with leaf development were randomly selected, including the photosynthesis pathway (unigene TR10048 and unigene TR19418), plant auxin signal pathway (unigene TR11070 and unigene TR14945), phenylpropanoid biosynthesis (unigene TR11546, unigene TR11244, unigene TR11203, and unigene TR10148), and flavonoid biosynthetic process (unigene TR8189 and unigene TR15898). As shown in Figure 6, these results of gene expression by RT-qPCR were identical to the differential expression that six libraries exhibited. Thus, it is useful to further investigate the genes related to leaf growth in *E. ulmoides*.

## 3. Discussion

The determination of physiological indices has great significance in studying the medicinal active ingredients of *E. ulmoides* and guiding production, it is a common index in plant research. Leaf area is closely related to photosynthetically active radiation and reflects the plant’s adaptability to factors such as its geographical environment and nutritional conditions [10]. Moreover, leaf area is an important physiological and morphological indicator that affects plant growth and development [11]; it is a more sensitive plant organ in response to UV-B radiation, and leaf area size can reflect the ability of leaves to receive light, which can increase the economic value of *E. ulmoides*. Most studies showed that the leaf area of plants decreases after enhanced UV-B radiation, while this study showed that the leaf area of plants increases after UV-B radiation. Sakalauskaite et al. [12] reported an increase in the leaf area of basil leaves after sUV-B radiation, suggesting that the increase in leaf area might contribute to the increase in biomass of the leaves, which is consistent with the fact that in our study after sUV-B radiation. Singh et al. [13] reported that the number of branches of kidney beans increased after UV-B radiation compared with the control, which was consistent with the results of our study. Dotto [14] and Gonçalves [15] et al. suggested that the increase in leaf thickness might be due to the appearance of air spaces between the upper epidermis and the palisade tissue, as the additional air space could alter light reflection and promote the utilization of sunlight. Moreover, the increase in leaf thickness could result in utilizing additional CO_2_ to increase the rate of photosynthesis in comparison to the control, leading to better development of leaves and then promoting the increase in leaf dry weight and fresh weight. The number of leaves and relative water content decreased after sUV-B radiation, and the degree of growth of *E. ulmoides* was slightly reduced, probably due to the fact that the treatment with low doses of UV-B radiation inhibited the growth of *E. ulmoides* seedlings and made their stems grow thicker to achieve the effect of strong seedlings [7].

Photosynthesis, growth and development, and the physiological metabolism of plants cannot be achieved without energy produced by light. The effect of UV-B radiation on plant photosynthesis is multifaceted [16]. The photosynthesis of plants is related to leaf area, and in this study, the leaf area of *E. ulmoides* increased after sUV-B radiation, and the net photosynthetic rate and actual photosynthetic efficiency were enhanced, so that the moderate amount of UV-B radiation was beneficial to photosynthesis in *E. ulmoides*. The chloroplast structure in *E. ulmoides* was well improved, and the photosynthetic substances of plants were effectively accumulated, which could better adapt to sUV-B radiation. sUV-B radiation significantly increased chlorophyll content in *E. ulmoides* for this study. Similar results were also found by Arróniz-Crespo et al. [17] and Sakalauskaitė et al. [18] in their experiments, which showed that appropriate UV radiation could promote an increase in photosynthetic efficiency and a high chlorophyll content in the plant. Carotenoids, stomatal conductance, intercellular CO_2_ concentration, and transpiration rate decreased, suggesting that their contents after sUV-B radiation may be affected by other factors, such as external ambient temperature, humidity, and soil conditions [19,20].

Light is also one of the main factors affecting secondary metabolites in plants and can regulate a variety of metabolic signals and synthetic pathways in plants [21]. Ultraviolet radiation can increase the content of certain secondary metabolites of medicinal importance in plants [22]. Many studies have shown that the pharmacological effects of medicinal plants are related to the content of their secondary metabolites [23]. In this study, the content of chlorogenic acid and genipin increased after sUV-B radiation, and there was a significant difference in the content of chlorogenic acid, which indicated that sUV-B radiation facilitated the accumulation of active ingredients in *E. ulmoides*. Chlorogenic acid belongs to phenols, and genipin belongs to terpenoids. Because phenols and terpenoids play an important role in scavenging free radicals and antioxidation, the increase in their contents after UV-B radiation may be related to their defensive effects in *E. ulmoides* [24]. It has been shown that phenolic compounds are important kinds of plant secondary metabolites that have a variety of pharmacological activities. Phenolic substances are mainly distributed in the upper epidermis, epidermis, waxy layer, and vacuole of plants and are sensitive to ultraviolet light, protecting plants like a natural UV filter. A large number of studies have proven that ultraviolet rays could induce many plants to produce phenolic compounds, which could shield UV-B radiation, help repair damaged DNA, improve plant antioxidant capacity, and affect the decomposition of substances [25,26]. On the contrary, the content of aucubin, geniposide, and geniposidic acid was lower than in the control group, indicating that sUV-B radiation inhibited the synthesis of aucubin, geniposide, and geniposidic acid. Aucubin, geniposide, and geniposidic acid belong to terpene glycosides, and the effect of UV-B on them has also been reported. Dolzhenko et al. [27] studied the effect of sUV-B radiation on the composition of *Mentha piperita* L. and showed that sUV-B radiation altered gene expression, enzyme activity, and the accumulation of defense metabolites. RT-qPCR analysis showed that this effect also included terpenoid biosynthesis and expression of coding genes, and it was concluded that terpenoids and flavonoids could be interconverted after UV-B regulation, thus causing a decrease in terpenoid content after sUV-B radiation. There are more metabolic pathways involved in aucubin, geniposide, and geniposidic acid in *E. ulmoides,* and the mechanism of its response to ultraviolet radiation needs further study.

Plant photomorphogenic responses were induced after sUV-B radiation, and appear to be adaptive, such as the rapid induction of a variety of gene expressions, including those involved in plant hormone metabolism and phenylpropanoid synthesis [28,29,30]. Our Kyoto Encyclopedia of Genes and Genomes (KEGG) pathway enrichment analysis of the differentially expressed genes (DEGs) focused on these two aspects. In plant hormone signaling, related genes from the growth hormone response proteins aux/iaa, growth hormone response protein iaa, and Saur family proteins are upregulated, and high expression of cre 1 (EC:2.7.13.3) is involved in plant biosynthesis metabolism, which promotes plant cell division and accelerates the growth of plant leaves. Ido Shwartz et al. [31] specifically emphasized the balance of plant hormones between morphogenesis and differentiation during leaf development. Auxin plays a role in nearly all plant developmental processes, and leaf development is no exception. Thus, we speculated that the increase in gene expression of auxin signaling components could accelerate leaf growth and increase leaf area in our study. Furthermore, lignins mainly provide strength and rigidity to the cell wall and are also involved in defense mechanisms against biotic and abiotic stresses [32,33,34]. In our study, the upregulation of the gene encoding peroxidase [EC:1.11.1.7] could improve lignin metabolic biosynthesis, which might change the structure of plant cell walls, increase leaf thickness, and enhance the resistance abilities of plants after sUV-B treatment. Lignin biosynthesis through the phenylpropanoid pathway encompasses a complex network [35]. The activity of peroxidase (POD) in lignin biosynthesis catalyzes the polymerization of lignin monomer as the last step of lignin synthesis [36,37]. In this study, the genes related to auxin metabolism are all up-regulated, which explains the reasons for the rapid growth of plants, on the other hand. At the same time as the up-regulation of plant hormones, we also found that the genes related to key enzymes regulating lignin metabolism showed an upward trend, which promoted lignin synthesis. The increase in lignin can promote the increase in plant cell wall thickness, and the research results are consistent with the results of leaf tissue sections, which explains the reason for the cell wall thickening of mesophyll cells. Phenylpropanoids are the prerequisite substances for chlorogenic acid synthesis. sUV-B radiation promoted the expression of the genes related to phenylpropanoids and increased the content of chlorogenic acid, a secondary metabolite in *E. ulmoides*. It is helpful to provide a good method for popularizing the planting of ULMEP in northeast China.

## 4. Materials and Methods

### 4.1. Test Materials and Treatment Conditions

This experiment was conducted outdoors at Northeast Forestry University, Heilongjiang Province, China (45.75° N, 126.63° E, 1826 m above sea level) for the culture of leafy *E. ulmoides* seedlings (the growing season was from 25 June 2016 to 25 June 2017). Five-year-old *E. ulmoides* seedlings of uniform growth were transplanted into pots (25 cm in diameter and 35 cm deep), one seedling per pot. The substrate used for seedling growth was peat: vermiculite: perlite (1 V:1 V:1 V), and incubation was carried out at 26 °C under natural light. The experiment consisted of two treatments: (1) natural light solar radiation (control, CK) and (2) artificially applied ambient UV-B (treatment group, T) with a radiation intensity of 2.81 KJ·m^−2^·d^−1^ for 3 days, with three replicates set up. After one year of recovery growth, the collection of leaves from the control and treatment groups was started on 25 June 2017, for the purpose of performing each index measurement.

### 4.2. Measurement of Physiological Indexes of E. ulmoides

#### 4.2.1. Number of Branches, Number of Leaves, Leaf Length, Leaf Width, Leaf Area and Thickness Statistics of *E. ulmoides*

The number of uniaxial branches of *E. ulmoides* was counted, and the number of leaves was obtained from the count of each plant. Leaf length, leaf width, and leaf area of each seedling were obtained by the grid method [38] measurements. Leaf thickness was measured using a semi-thin cross-section cut from the leaf material using a micrometer (7331MEXRL-25 model, Jiangsu China).

#### 4.2.2. Leaf Anatomy

Leaf samples for trans-section examinations were cut into pieces and fixed in FAA solution. Fixed samples were dehydrated in a graded ethanol series, immersed in a graded ethanol series, and embedded in paraffin for sectioning; transverse sections were made on a Leica 2016 rotary microtome (Leica Inc., Bensheim, Wetzlar, Germany), mounted on glass slides, treated with a safranin and fast-green staining procedure, and dehydrated through an ethanolic series. Sections were observed with a Nikon (TE2000-U, Tokyo, Japan) electron microscope equipped with a 20 × 0.45 objective lens, and images were obtained.

#### 4.2.3. Measurement of Dry Weight and Fresh Weight of Leaves

(1) Fresh weight determination: cut the leaves of *E. ulmoides* into a container (or plastic bag) of known weight, bring it into the room, and weight the fresh weight (FW) with an analytical balance (FA2004N model, Shanghai China).

(2) Dry weight determination: open the oven in advance and raise the temperature to 100–105 °C. Put the weighed fresh weight of *E. ulmoides* and *E. ucommia* leaf material into a paper bag, put it into the oven, preheat it for 10 min at 100–105 °C, then lower the oven temperature to 70–80 °C, and bake it to a constant weight. Remove the paper bag and material, put them into the desiccator to cool to room temperature, and weigh the dry weight (DW). After obtaining the above data, calculate the relative water content according to the following formula:
$$\mathrm{Relative}\;\mathrm{water}\;\mathrm{content}\;(\%)=\frac{\mathrm{FW}-\mathrm{DW}}{\mathrm{FW}}*100$$

#### 4.2.4. Measurement of Photosynthetic Parameters

Selection of leaves that were fully expanded, in good growing condition, and uniformly leafed. And under the condition of an environmental temperature of 25 °C, the net photosynthetic rate (Pn, μmol·m^−2^·s^−1^), stomatal conductance (Cond, mol·m^−2^·s^−1^), intercellular CO_2_ concentration (Ci, μmol·mol^−1^), transpiration rate (Tr, mmol·m^−2^·s^−1^) were measured by portable photo-synthesizers (Li-6400 model, New York, NY, USA).

#### 4.2.5. Measurement of Photosynthetic Pigments

The content of photosynthetic pigments was determined by referring to the method of Jiang et al. [39] with slight modifications. Firstly, 0.2 g of fresh *E. ulmoides* leaves were accurately weighed, added to 80% pre-cooled acetone, ground into a homogenate, and centrifuged at 6000 rpm for 15 min. The supernatant was aspirated to determine the absorbance values at 665, 649, and 480 nm, and the experiment was kept in a dark environment throughout. The mass fractions of chlorophyll a (Chl a) and chlorophyll b (Chl b) were calculated using the following formula:Chl a = 12.19 × A_665_ − 3.45 × A_649_
Chl b = 21.99 × A_649_ − 5.32 × A_665_
Chl = Chl a + Chl b
Car = (1000 × A_480_ − 2.14 × Chl a − 70.16 × Chl b)/220
Chl a/b = Chl a/Chl b

#### 4.2.6. Measurement of Chlorophyll Fluorescence Parameters

The chlorophyll fluorescence parameters were determined by portable the PAM-2500 chlorophyll fluorescence instrument, referring to the method of Chen Yanger et al. [40]. The leaves needed to be fully dark-adapted for 20 min, and the minimum fluorescence (F_0_) after dark adaptation was recorded by turning on the measurement light, and the maximum fluorescence (Fm) after dark adaptation was recorded by turning on the saturation pulse light (8000 µmol·m^−2^·s^−1^, 0.6 s) and then turning off. The maximum fluorescence Fm’ was recorded by turning on the action light (600 µmol·m^−2^·s^−1^), recording the photosynthetic steady-state fluorescence (Fs) after the leaf photosynthesis reached the steady state, turning on the saturating pulsed light (8000 µmol·m^−2^·s^−1^, 0.6 s), turning off the action light, turning on the far-red light, and turning off after about 8 s to obtain the minimum fluorescence F_0_′. The maximum photosynthetic efficiency and the actual photosynthetic efficiency were calculated with the following formula:Fv/Fm = (Fm − F_0_)/Fm
Y(II) = (Fm’ − Fs’)/Fm’

### 4.3. Determination of Secondary Metabolite Content of E. ulmoides

The extraction of the major secondary metabolites in *E. ulmoides* leaves was carried out using the flash extraction method with reference to the method of Zhang Qiancheng [41]. Thirty grams of each *E. ulmoides* leaf was taken from the control group and the treatment group after low-temperature drying, chopped, and put into the flash extractor. Then, a 70% ethanol solution was added to the flash extractor (JHBE-50A model), using 110 V for flash extraction. The extraction was performed every 5 min, during which the apparatus was suspended for 30 s every 1 min, and the filtrate after extraction was coarsely filtered with medicinal gauze. The above operation was repeated three times, and all the filtrates were combined, and the test solution was obtained after fine filtration. The content of the five active ingredients (aucubin, chlorogenic acid, genipin, geniposide, and geniposidic acid) was determined simultaneously by HPLC under the following chromatographic conditions: methanol as mobile phase A and phosphoric acid (0.5%) aqueous solution as mobile phase B; the flow rate was 1.0 mL·min^−1^. The detection wavelengths were 206 nm from 0 to 15 min, 236 nm from 15 to 55 min, and the detection wavelengths were 1.0 mL·min^−1^. The gradient elution: 5%→10% A, 95%→90% B, 0–30 min; 10%→25% A, 90%→75% B, 30–70 min. Finally, the extraction rate of the active ingredients of *E. ulmoides* was calculated.

### 4.4. cDNA Library Preparation and Illumina Sequencing for Transcriptome Analysis

Total RNA from the samples was extracted using TRIzol (Sangon, Shanghai, China) following the manufacturer’s instructions and then treated with RNase-free DNase I. Preparation of the cDNA library was described in detail in a previous study [28]. Six cDNA libraries were constructed from the leaves of the control and sUV-B treatments, in which equal amounts of total RNA were pooled from three biological replicates. The six libraries were used for a comparative analysis of transcriptome sequencing. Finally, six libraries were sequenced at the Beijing ORI-gene Institute (ORI, Beijing, China), and reads were generated in a 100 bp paired-end format according to the manufacturer’s instructions (Illumina Inc., San Diego, CA, USA). All raw transcriptome data were deposited in the GenBank Short Read Archive.

### 4.5. Denovo Assembly and Functional Annotation Analysis of Illumina Sequencing

Raw reads from the sequencing platform were generated by base calling, referring to Conesea’s method [42]. These contigs were then further processed with sequence clustering software, TGICL- 2.1.tar.gz, to form longer sequences defined as unigenes. The generated unigenes were used for BLASTX alignment (E < 0.00001) and annotation against protein databases, including nonredundant (NR), SwissProt, Clusters of Orthologous Groups for Eukaryotic Complete Genomes (KOG), and Kyoto Encyclopedia of Genes and Genomes (KEGG) protein databases. With NR annotation, the Gene Ontology (GO) annotation of unigenes was obtained by using the Blast2GO program [43], and GO functional classification for all unigenes was performed by WEGO 2.0 software.

### 4.6. Identification of Differentially Expressed Genes (DEGs)

To compare the differences in gene expression in leaves at the control and sUV-B radiation treatments, the RPKM method (reads per kb per million reads) was used to calculate read density. The FDR (false discovery rate) was used to determine the threshold *p*-value in multiple tests. According to Zhang’s [42] approach, we used an FDR < 0.01, *p* ≤ 0.05, and an absolute value of the log2 ratio > 1 as the threshold to determine significant differences in gene expression. The DEGs were used for GO and KEGG enrichment analyses according to Zhang’s method [44].

### 4.7. Quantitative Real-Time PCR Validation

Total RNA was extracted as described above. Total RNA (2 mg) was subjected to reverse transcriptase reactions using the RevertAid H Minus First-Strand cDNA Synthesis Kit (MBI Fermentas, Burlington, ON, Canada) according to the manufacturer’s instructions. The sequences of the specific primers are listed in Additional File 1. The constitutively expressed gene ribosomal 40S protein S9 (rps9, AJ749993.1) was used as the internal control. The first-strand cDNA was diluted 50 times with nuclease-free water and used in RT-qPCR. Each reaction contained a mixture of 1.5 μL of diluted cDNA aliquots, 1.5 μL of mixed primers, 10 μL of BeyoFast™ SYBR Green qPCR Mix (Beyotime, Nanjing, China), and 6.0 µL of nuclease-free water. RT-qPCR amplification was carried out in 96-well plates on a LightCycler^®^ 480II System (Roche, Switzerland; Roche Diagnostics, Indianapolis, IN, USA). The expression level was calculated with 2^−ΔΔCt^ and normalized to the Ct value of rps9. The qRT-PCR results were obtained from three biological replicates and three technical repeats for each gene and sample.

### 4.8. Data Analysis

The experimental data were analyzed and processed using SPSS 19.0 software, Origin 2022, and Microsoft Excel 2010.

## 5. Conclusions

The effects of a moderate amount of sUV-B radiation on *E. ulmoides* during its growth were manifold. The experimental results showed that sUV-B radiation increased the leaf area, leaf thickness, number of branches, leaf fresh weight, and leaf dry weight. After sUV-B radiation, the contents of chlorophyll a and chlorophyll b increased, the net photosynthetic rate increased, the transpiration rate decreased, and the actual photosynthetic efficiency increased, which promoted the contents of some secondary metabolites and was beneficial to the accumulation of genipin and chlorogenic acid. At the same time, it also showed that the content of the secondary metabolites chlorogenic acid and genipin was related to plant growth parameters. The leaves were the basis for the accumulation and changes of chemical components, and the increase in leaf area and its photosynthesis were also enhanced, which is favorable to the synthesis of the active components genipin and chlorogenic acid. Two transcriptome databases were integrated, and the differentially expressed genes (DEGs) from plant auxin and phenylpropanoid biosynthesis signal pathways were focused on after sUV-B treatment. Up-regulated single genes participate in plant hormone signal transduction and phenylpropanoid biosynthesis metabolism through the KEGG pathway, which further affects the physiological indexes of *E. ulmoides* and the content of chlorogenic acid, a secondary metabolite. Photosynthetic-related genes, starch and sucrose metabolism, nitrogen metabolism, ABC transporter, oxidative phosphorylation, plant hormones, and other metabolic pathways of *E. ulmoides*. With the improvement in the overall metabolic level of *E. ulmoides*, plants have more energy accumulation; that is, more “sources” lead to an increase in “sinks”, which eventually leads to stronger and faster growth of plants. In this study, the genes related to auxin metabolism are all up-regulated, which explains the reasons for the rapid growth of plants, on the other hand. In this study, the response mechanism of *E. ulmoides* induced by ultraviolet rays was preliminarily constructed through the interaction between different genes, and the key node genes were screened out. The expression results of qRT-PCR from 10 candidate unigenes were similar to those from transcriptome database analysis. The results provide theoretical guidance for promoting the growth and development of *E. ulmoides*, producing medicinal active components, and improving the utilization rate of *E. ulmoides*. It also shows that sUV-B radiation is one of the most important factors popularizing the planting of *E. ulmoides* using the leaf model in Heilongjiang Province. Furthermore, it is conducive to protecting tree species from damage.

## Figures and Tables

**Figure 1 ijms-24-08168-f001:**
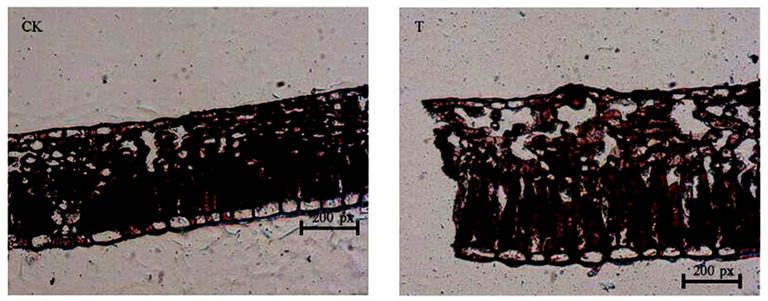
Anatomical characteristics of *E. ulmoides*; CK: anatomical characteristics of leaves under natural light (bar = 200 px); T: anatomical characteristics of leaves after UV-B treatment (bar = 200 px).

**Figure 2 ijms-24-08168-f002:**
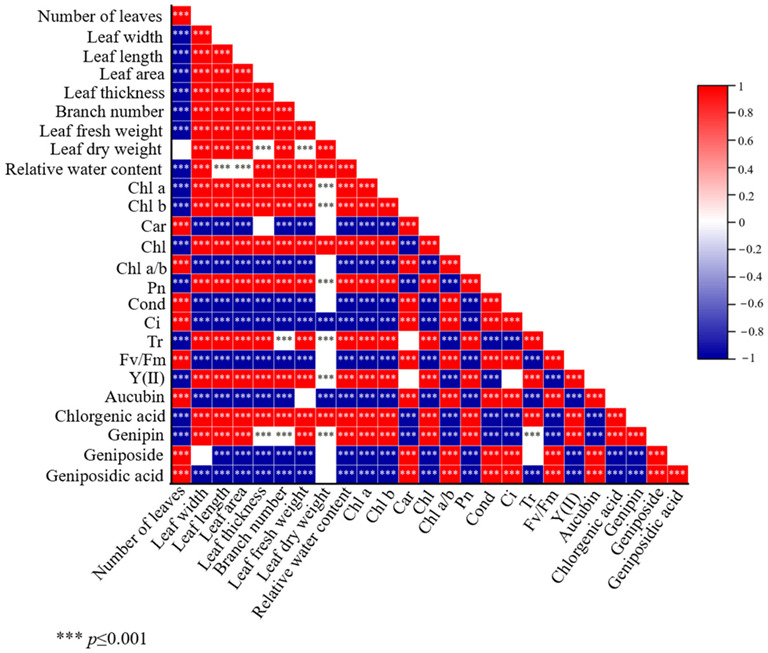
Heat map of correlations (indicates significant differences between indicators; red indicates strong positive correlation; blue indicates strong negative correlation).

**Figure 3 ijms-24-08168-f003:**
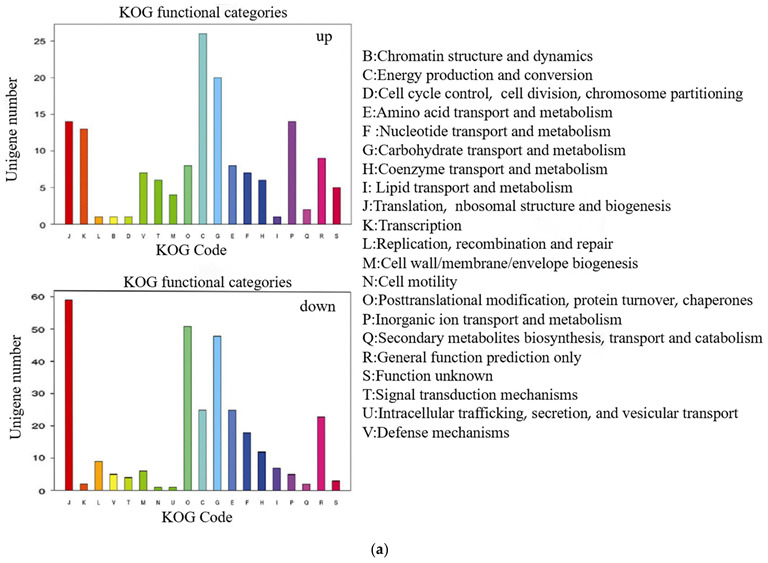
Number distribution and pathway analysis of differentially expressed genes. (**a**) Number distribution of unigenes assigned to the COG database. *X*-axis: COG functional classification. *Y*-axis: numbers of unigenes assigned to different COG functional classifications. (**b**) DEGs KEGG taxonomy. DEGs related to photosynthesis pathway, plant auxin, and phenylpropanoid biosynthesis. (**c**) Percent and number distributions of unigenes assigned to the GO database in the UV-B-treated leaves. *X*-axis: GO categories. *Y*-axis: number of unigenes assigned to different GO categories.

**Figure 4 ijms-24-08168-f004:**
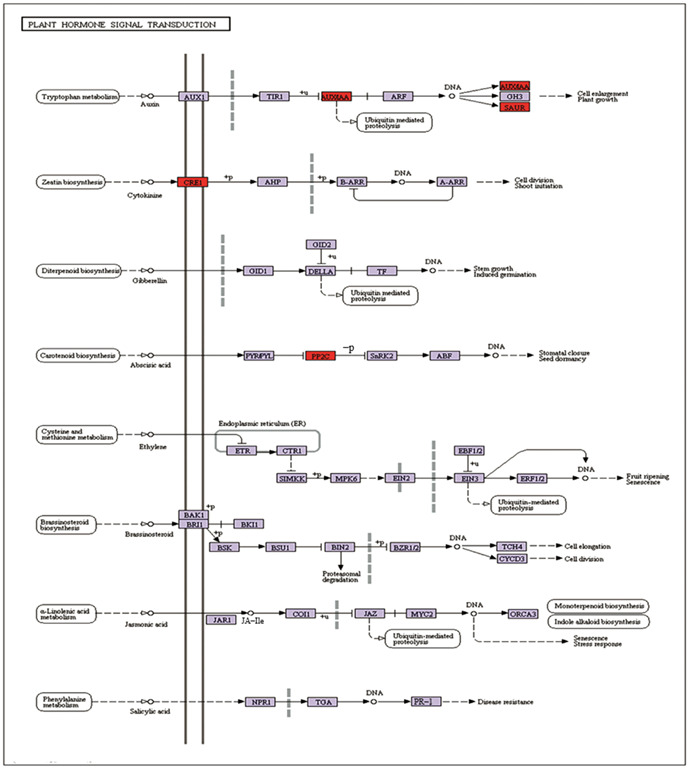
Network analysis of various auxins involved in the regulation of the leaf development process. Red indicates the key up-regulated genes associated with the auxin signaling pathway, purple indicates down-regulated genes, and white indicates both up-regulated genes and down-regulated genes.

**Figure 5 ijms-24-08168-f005:**
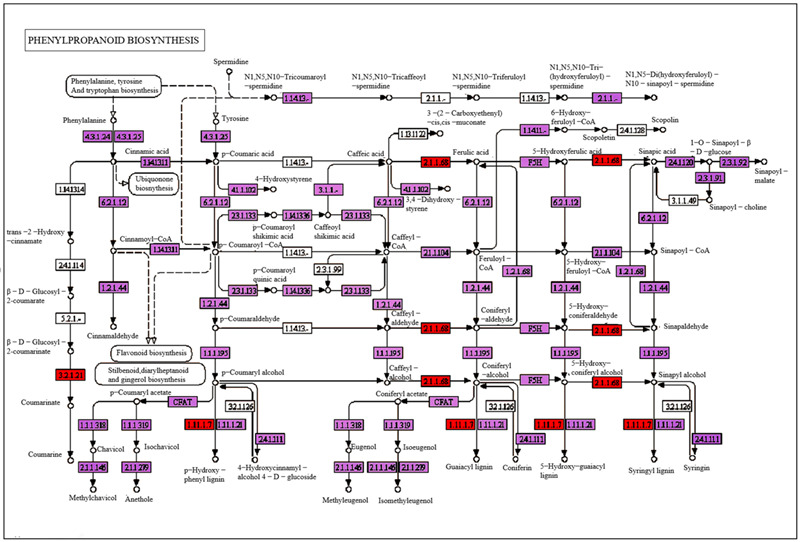
The putative phenylpropanoid biosynthesis metabolic pathway of *E. ulmoides* was constructed based on KEGG annotation. These unigenes are distributed in the rectangular boxes in the figure. Red indicates the key up-regulated genes associated with the lignin metabolic pathway, purple indicates down-regulated genes, and white indicates both up-regulated genes and down-regulated genes.

**Figure 6 ijms-24-08168-f006:**
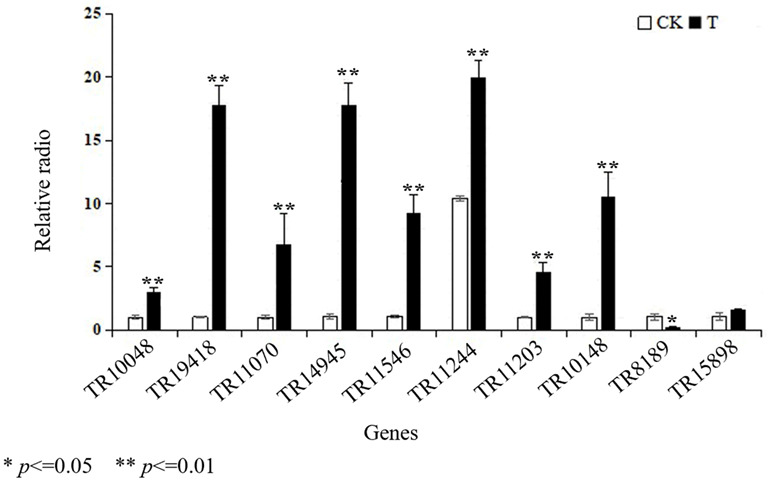
The 10 candidate unigenes associated with leaf development were detected under the two treatments. (Ribosomal 40S protein S9 as an internal control and CK as the control).

**Table 1 ijms-24-08168-t001:** The size of CK in epidermal cells and palisade tissue of *E. ulmoides* leaves.

Samples	Epidermal Cell (px)	Palisade Tissue (px)
CK	80.00 ± 4 ^b^	145.45 ± 7.27 ^b^
T	166.67 ± 8.33 ^a^	364.64 ± 18.23 ^a^

Note: CK was the control and T was the sUV-B radiation intensity of 2.81 kJ m^−2^ d^−1^; a and b stand for significance, different lowercase letters in the same column represent a significant difference (*p* < 0.05), and the same lowercase letters in the same column represent no significant difference (*p* > 0.05).

**Table 2 ijms-24-08168-t002:** Changes in growth indexes of *E. ulmoides*.

Treatment	Number of Leaves (Piece)	Leaf Width (cm)	Leaf Length (cm)	Leaf Area (cm^2^)	Leaf Thickness (mm)	Branch Number (Branch)	Leaf Fresh Weight (g)	Leaf Dry Weight (g)	Relative Water Content (%)
CK	139.00 ± 6.95 ^a^	6.62 ± 0.33 ^a^	12.59 ± 0.63 ^b^	61 ± 3.05 ^b^	0.23 ± 0.012 ^b^	11 ± 1.00 ^b^	163.77 ± 8.19 ^b^	57.39 ± 2.87 ^b^	0.64 ± 0.008 ^a^
T	132.00 ± 6.6 ^a^	9.24 ± 0.46 ^a^	17.32 ± 0.87 ^a^	121 ± 6.05 ^a^	0.31 ± 0.016 ^a^	32 ± 1.15 ^a^	247.12 ± 12.36 ^a^	92.84 ± 4.64 ^a^	0.62 ± 0.006 ^a^

Note: CK was the control and T was the sUV-B radiation intensity of 2.81 kJ m^−2^ d^−1^; a and b stand for significance, different lowercase letters in the same column represent a significant difference (*p* ≤ 0.05), and the same lowercase letters in the same column represent no significant difference (*p* > 0.05).

**Table 3 ijms-24-08168-t003:** Changes in photosynthetic pigment indexes of *E. ulmoides*.

Treatment	Chlorophyll a (Chl a)	Chlorophyll b (Chl b)	Carotenoid (Car)	Chlorophyll (Chl)	Chlorophyll Ratio (Chl a/b)
CK	0.89 ± 0.055 ^a^	0.47 ± 0.025 ^a^	0.082 ± 0.006 ^a^	1.36 ± 0.079 ^a^	1.87 ± 0.028 ^a^
T	0.91 ± 0.023 ^a^	0.49 ± 0.013 ^a^	0.069 ± 0.007 ^b^	1.405 ± 0.036 ^a^	1.85 ± 0.036 ^b^

Note: CK was the control and T was the sUV-B radiation intensity of 2.81 kJ m^−2^ d^−1^; a and b stand for significance, different lowercase letters in the same column represent a significant difference (*p* ≤ 0.05), and the same lowercase letters in the same column represent no significant difference (*p* > 0.05).

**Table 4 ijms-24-08168-t004:** Changes in fluorescence parameters of *E. ulmoides*.

Treatment	Net Photosynthetic Rate (Pn)	Stomatal Conductance (Cond)	Intercellular CO_2_ Concentration (Ci)	Transpiration Rate (Tr)	Maximum Light and Efficiency (Fv/Fm)	Actual Photosynthetic Rate(Y(II))
CK	13.34 ± 0.63 ^a^	0.27 ± 0.033 ^a^	281.8 ± 12.14 ^a^	4.53 ± 0.32 ^a^	0.71 ± 0.073 ^a^	0.47 ± 0.04 ^b^
T	14.03 ± 1.46 ^a^	0.25 ± 0.037 ^a^	273.89 ± 21.6 ^a^	6.13 ± 0.19 ^a^	0.69 ± 0.043 ^a^	0.52 ± 0.057 ^a^

Note: CK was the control and T was the sUV-B radiation intensity of 2.81 kJ m^−2^ d^−1^; a and b stand for significance, different lowercase letters in the same column represent a significant difference (*p* ≤ 0.05), and the same lowercase letters in the same column represent no significant difference (*p* > 0.05).

**Table 5 ijms-24-08168-t005:** Changes in secondary metabolites in *E. ulmoides*.

Treatment/Secondary Metabolites	Aucubin (ug/g)	Chlorogenic Acid (ug/g)	Genipin (ug/g)	Geniposide (ug/g)	Geniposidic Acid (ug/g)
CK	68.85 ± 0.004 ^a^	47.46 ± 0.008 ^b^	0.31 ± 0.005 ^a^	39.84 ± 0.213 ^a^	121.84 ± 0.008 ^a^
T	4.6857 ± 0.005 ^b^	83.0679 ± 2.98 ^a^	0.35 ± 0.004 ^a^	9.15 ± 0.002 ^b^	4.85 ± 0.045 ^b^

Note: CK was the control and T was the sUV-B radiation intensity of 2.81 kJ m^−2^ d^−1^; a and b stand for significance, different lowercase letters in the same column represent a significant difference (*p* ≤ 0.05), and the same lowercase letters in the same column represent no significant difference (*p* > 0.05).

**Table 6 ijms-24-08168-t006:** The number of de novo assembly.

	All (≥200 bp)	≥500 bp	≥1000 bp	N50	GC (%)	Total Length	Max Length	Min Length	Average Length
Transcript	62,287	40,111	25,753	1762	42	70,449,627	14,647	224	1131
Unigene	42,333	23,627	14,132	1582	42.22	41,056,860	14,647	224	969

**Table 7 ijms-24-08168-t007:** The number of unigenes annotated to different databases.

Unigene	SwissProt	TrEMBL	NR	Pfam	KOG	GO	KO
42,333	14,988	24,272	24,500	19,708	19,523	19,282	6790
100%	35.40%	57.30%	57.90%	46.60%	46.10%	45.50%	16.00%

## Data Availability

All data are contained within the article.

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
