# Peer review of "Supplementary UV-B Radiation Effects on Photosynthetic Characteristics and Important Secondary Metabolites in Eucommia ulmoides Leaves"

_ijms, 2023, doi:10.3390/ijms24098168_

Round 1
Reviewer 1 Report
The article is devoted to the effect of additional UV-B radiation on photosynthetic characteristics and important secondary metabolites in the leaves of Eucommia ulmoides. This material is quite interesting and relevant, however, I have a few comments.
1. Figure 1 is of insufficient quality (images), and the table below the figure must be added to the text of the article.
2. How can the authors explain the fact that the mass of the leaf has increased, and the water content in them has decreased?
3. It seems to me that it would be better to divide Figure 2 into 2 separate figures, because in this form Figure 2b is poorly readable.
4. Figure 5 is also not very good quality, please edit it.
Author Response
Point 1: Figure 1 is of insufficient quality (images), and the table below the figure must be added to the text of the article.
Response 1: We have improved the quality of pictures and added tables to the body of the article.
Point 2: How can the authors explain the fact that the mass of the leaf has increased, and the water content in them has decreased?
Response 2: The fresh weight and dry weight of leaves increased at the same time, and the fresh weight increased even more. Formula: Water content is equal to (fresh weight-dry weight)/fresh weight. Because the fresh weight decreased more than the dry weight, the water content decreases less, and there was no significant difference between the control and treatment groups.
Point 3: It seems to me that it would be better to divide Figure 2 into 2 separate figures, because in this form Figure 2b is poorly readable.
Response 3: We divided Figure 2 into two diagrams, which improved the readability of Figure 2b.
Point 4: Figure 5 is also not very good quality, please edit it.
Response 4: We have improved the clarity of Figure 5.
Reviewer 2 Report
There are plenty of papers showed that applications of UV-B will bring both damage and protection for plants, This study by Xiao et al. evaluated the effects of ultraviolet light supplemental on the leave of the important medicinal plant, E.ulmoides. it would provide some benefits for further applications of UV-B on the similar plants.
The manuscript is okay to read. However, the manuscript is not well written. The writing needs to be improved. Many small errors are scattered throughout the manuscript. I recommend a thorough check of the grammar/ formatting. Comments are listed below.
Major comments:
1) The introduction part is too short. This manuscript is mainly focus on the benefit of sUV-B, However, it well known that supplementary of UV-B will cause damage for plants. It depends on the treatment conditions, such as UV dose, timing, material growth stage and the different time point of the material collected after the treatment. Suggest the author to add a short introduction of the damage effects by sUV-B.
2) Could the author explain why choose the treatment of “low-dose UV-B radiation (2.81 KJ·m-2 ·d-1 )” for 3 days and then collected leaves after one year of recovery growth? Does the author have some data record for leaf area at different time point such as 7d, 14d, 1 month after the treatment?
3) Could the author explain that why they used two CDNA library? Why not six CDNA library, three for WT, three for UV-B treated?
4) The text labeling in figure 1, 3, 6 are unclear, need to improve the quality. Figure 6, please adjust the position of the label ”*” in the figure. Please type it, It can clearly see some background from copies. Please explain the mean error refer to SD or SE? and ”*” the statistical significance of differences in the figure 6 legend.
5) The conclusion part in Line 348, ”It is helpful to provide a good method for popularizing the planting the ULMEP in northeast China.” Please tone down the statement. Although the author found some benefit by sUV-B, but the author cannot exclude the side effects.
Besides that, there are lots of grammar/ formatting errors, a few examples listed below, please check the similar errors very carefully throughout the manuscript.
1) E.ulmoides in the paragraph from line 86-96, line 364, should be in italic.
2) Formatting/Font style error: Line 17 “Oliver(E.ulmoides), the effects of”, Line 153 “respectively (Table 5).”
3) The use of “space” error, Line 260: “beans increasedafter UV-B”;
Line 69, ”n (2.81 KJ·m-2 ·d-1 )as”;
line 427 “30g”;
line 384 “10min”;
line 431 “30s”;
line 148, “(b)Heat”.
4) Line 270 ”the growth of Eucommia seedlings”, line 383 “weight of Eucommia leaf”
5) Line 290, “Many studies have shown that the pharmacological effects of medicinal plants were related to the content of their secondary metabolites.” Need to cite related references.
Round 2
Reviewer 1 Report
I thank the authors for their attentive attitude to the comments.
Unfortunately, the quality of figures 2b and 5 is still not suitable for publication. Please replace them in the same way as picture 3.
Author Response
Point 1: The quality of figures 2b and 5 is still not suitable for publication. Please replace them in the same way as picture 3.
Response 1: We have improved the quality of Figure 2b and Figure 5.
Reviewer 2 Report
The authors have addressed most of my comments. I do not have any major comments, but some minor comments. The authors need to improve the quality of the figures.
There are some background in Figure 3b and Figure 6, Please check it very carefully.
As to Figure 2 and 6, suggest the authors to type the text to improve the quality of the figures by Adobe Illustrator or similar software.
Author Response
Point 1: There are some background in Figure 3b and Figure 6, Please check it very carefully.
As to Figure 2 and 6, suggest the authors to type the text to improve the quality of the figures by Adobe Illustrator or similar software.
Response 1: We have removed some backgrounds in Figure 3b and Figure 6, and improved the graphic quality of Figure 2 and Figure 6.